# Vitamin K2 Holds Promise for Alzheimer’s Prevention and Treatment

**DOI:** 10.3390/nu13072206

**Published:** 2021-06-27

**Authors:** Alexander Popescu, Monica German

**Affiliations:** 1Undergraduate Program, Harvard Extension School, Cambridge, MA 02138, USA; apopescu@mit.edu; 2Department of Clinical Medicine, Pacific Northwest University of Health Sciences, Yakima, WA 98901, USA

**Keywords:** vitamin K2, menaquinone, Alzheimer’s disease, dementia, neurodegeneration, cognition, neuroinflammation, microglia, antioxidant, microbiome

## Abstract

Recent studies have highlighted the importance of vitamin K2 (VK2) in human health. However, there have been no clinical studies investigating the role of VK2 in the prevention or treatment of Alzheimer’s disease (AD), a debilitating disease for which currently there is no cure. In reviewing basic science research and clinical studies that have connected VK2 to factors involved in AD pathogenesis, we have found a growing body of evidence demonstrating that VK2 has the potential to slow the progression of AD and contribute to its prevention. In our review, we consider the antiapoptotic and antioxidant effects of VK2 and its impact on neuroinflammation, mitochondrial dysfunction, cognition, cardiovascular health, and comorbidities in AD. We also examine the link between dysbiosis and VK2 in the context of the microbiome’s role in AD pathogenesis. Our review is the first to consider the physiological roles of VK2 in the context of AD, and, given the recent shift in AD research toward nonpharmacological interventions, our findings emphasize the timeliness and need for clinical studies involving VK2.

## 1. Introduction

The incidence of Alzheimer’s disease (AD) has risen considerably in recent years, and AD remains a leading cause of chronic disability and death. As the most common type of dementia, AD affects an estimated 6.2 million Americans, a number that is projected to more than double by 2050 [1]. In 2018, there were 50 million people worldwide living with dementia [2]. The emotional, physical, and financial toll of AD impacts not only individuals and families, but also society more broadly, and there is no treatment to cure it or to slow its progression.

The two hallmarks of AD are the extracellular deposits of neurotoxic β-amyloid (Aβ) in the brain and the accumulation of tau protein tangles in the neurons. Additionally, a variety of genetic and non-genetic factors have been implicated in its pathogenesis. One example of a well-established genetic factor is the alleles of the apolipoprotein E (*APOE*) gene. Having the *e4* allele (and particularly having two copies) increases AD risk, while having the *e2* allele partially protects against the disease [3]. In some cases of familial AD, a rare form of the disease, there are mutations in the presenilin 1 (*PSEN1*) and β-amyloid precursor protein (*APP*) genes [4]. The most important non-genetic risk factors for late-onset AD are aging, type II diabetes, cardiovascular disease, traumatic brain injury, sleep disturbances, poor diet, and a lack of exercise [5,6]. Among the factors implicated in AD pathogenesis are neuroinflammation, oxidative stress, mitochondrial dysfunction, vascular disease, and microbiome changes, all of which we will discuss later.

We are interested in whether or not vitamin K2 (VK2) might attenuate the effects of these factors and thus decrease the likelihood of developing AD. Recent studies have shown that VK2 has many physiological roles, and we believe that it is of interest in AD research. We review basic science research and clinical studies, including population-based cohort studies and randomized controlled trials. One distinguishing criterion for our search was our interest in VK2 rather than vitamin K1 (VK1) or vitamin K in general. Although VK1 and VK2 are related, our research showed that there is a difference between the two and their relationship to human health. When our original PubMed query for “Menaquinone and Alzheimer” and “Vitamin K2 and Alzheimer” found only six studies relevant to our review, we expanded our search to investigate the link between VK2 and factors involved in AD pathogenesis (Figure 1). For example, we researched population studies that considered the connection between cardiovascular health and AD, followed by studies that linked cardiovascular health with VK2.

Our review is of importance because it highlights new, emerging research relating VK2 and AD, and, to the best of our knowledge, it is the first review to explore this connection. By drawing attention to this relationship, we hope to communicate the need for future studies exploring the role of VK2 in AD prevention.

## 2. Comparison of Vitamins K1 and K2

Vitamin K1 (VK1) and vitamin K2 (VK2) are the two natural forms of vitamin K, and they differ in their chemical structure, physiological roles, diet sources, and bioavailability (Figure 2). While VK1 exists in only one form, the menaquinone (MK) family can be subdivided into multiple isoforms, classified as follows: short chains (MK-4 through MK-6), intermediate chains (MK-7 through MK-9), and long chains (MK-10 through MK-13). The name for each MK isoform corresponds to the number of unsaturated five-carbon groups (prenyl units) it contains.

Commonly referred to as phylloquinone, VK1 is the plant form of vitamin K, and it is primarily found in green, leafy vegetables. Our interest in VK2 stems in part from the fact that the Western diet is low in this necessary nutrient. Women following a Western diet had a much lower VK2 intake (36 mcg compared to 154.1 mcg) than Japanese women who regularly consumed nattō, a traditional Japanese dish made from fermented soybeans [7,8]. On average, VK2 comprises just 10% of dietary vitamin K in European diets [9]. Almost all the MKs are synthesized by bacteria, and most can be found in fermented foods, such as nattō, sauerkraut, pickled vegetables, and some types of cheese. MK-4, the only VK2 isoform that is obtained through conversion from VK1 and not from bacterial synthesis, is present in meat, fish, and egg yolk [10,11].

The recommended daily intake (RDI) of vitamin K is based exclusively on VK1, and we agree with the authors of a recent review who argue that, because of its unique physiological roles, VK2 warrants a separate RDI [12]. One additional consideration that we will discuss later is the contribution of intestinal VK2 synthesized by gut bacteria to the total physiological requirement for vitamin K. VK2 is better absorbed than VK1, and, of the different MKs, MK-7 is the most bioavailable [11,13].

Both VK1 and VK2 are cofactors of the γ-glutamyl carboxylase (GGCX) enzyme, which catalyzes a post-translational carboxylation that is essential for the activation of a special set of proteins called vitamin K-dependent proteins (VKDPs). The preferential target for VK1 is the liver, where it activates hepatic VKDP coagulation factors (II, VII, IX, and X) and coagulation proteins (C, S, and Z). Protein S, a hepatic VKDP, has been shown to have an essential role in neuronal protection from excitotoxic injury via the activation of the TAM receptor tyro3-phosphatidylinositol 3-kinase-Akt pathway [14]. One study demonstrated that protein S improved blood flow during ischemic brain injury, due to its antithrombotic effects, and directly protected neurons from hypoxic injury, leading to a better neurological outcome [15]. In support of those findings, another study determined that protein S had cytoprotective activities and preserved the integrity of the blood-brain barrier [16].

Studies suggest that VK1 may influence cognition. In older individuals, both dietary and serum VK1 were significant and independent predictors of cognitive function [17]. Lifetime consumption of a low VK1 diet resulted in cognitive deficits in rats [18,19]. Another study found a low intake of VK1 in nine of 31 elderly individuals who had early AD [20]. A cross-sectional study of 320 subjects between 60 and 75 years of age determined that there is a positive association between high serum levels of VK1 and verbal episodic memory, but not non-verbal episodic memory or executive functions [21]. Two other studies in elderly individuals (mean ages of 82 and 83, respectively) found a positive association between dietary intake of VK1 and memory questionnaire scores [22,23]. We note that many studies looking at VK1 and cognition had limitations in accurately assessing dietary VK1 [24]. One study did not find an association between low vitamin K levels and cognitive decline [25]. However, this study, unlike the others we have reviewed, examined individuals that had a mean age of only 60 and investigated the synergistic effects of vitamin K and vitamin D3.

There are also studies that discuss a possible association between the use of vitamin K antagonists (VKAs) and cognitive decline or brain focal atrophies [24]. These studies generally agree that anticoagulants impact cognition, with vitamin K as a possible mechanism. Given that VK2 also plays a role in coagulation, these studies are relevant to the effects of both VK1 and VK2 on brain health. One large cohort study treated subjects with VKAs and reported a decrease in visual memory and verbal fluency, but no changes in mini-mental state exam (MMSE) scores [26]. A cross-sectional study involving 54 patients, a third of whom were on VKAs, showed that focal brain atrophy, measured by MRI scans, was associated with the duration of treatment with VKAs [27]. Another study with subjects (≥75 years) on VKAs demonstrated that there is an association between VK1 concentration and cognition, which was assessed using the Milan Overall Dementia Assessment (MODA) [28]. A cohort study of 267 geriatric individuals found that the use of VKAs was associated with lower MMSE scores [29]. However, a cohort study of 378 elderly individuals did not find a difference in MMSE scores, although there was a change in frontal assessment battery (FAB) scores [30].

Both VK1 and VK2 have roles in brain health via the regulation of sphingolipid metabolism [19,31,32]. Sphingolipids are complex lipids that possess important biological functions in the development and survival of neurons. Alterations in sphingolipid metabolism have been implicated in neurodegenerative diseases such as AD, Parkinson’s disease, and Huntington’s disease [33]. In a study of mice that underwent demyelination to mimic multiple sclerosis lesions, an intake of 2 mg of VK1 three times per week increased the production of brain sulfatides after both one and three weeks of remyelination [34].

However, unlike VK1, VK2 activates a variety of extrahepatic VKDPs, which have many complex roles (Figure 3). Of these VKDPs, the best studied are the matrix Gla protein (MGP), osteocalcin, and growth arrest-specific protein 6 (Gas6). MGP inhibits the calcification of soft tissue and is present in the arterial wall [35]. Osteocalcin is important for bone health, and recent research has highlighted its additional roles in brain signaling, energy metabolism, and inflammation [36,37,38]. Gas6 is widely expressed in the nervous system, and it regulates neuroinflammation and cancer cell signaling [39,40].

VK2 also acts as a transcriptional regulator and has antioxidant properties [41,42,43]. Unlike VK1, VK2 possesses immunosuppressant effects and inhibits T-cell proliferation [44,45,46]. When coupled with research showing the role of VK2 in suppressing the NF-κB pathway, such findings provide convincing evidence of the immunomodulatory and anti-inflammatory properties of VK2 [47,48]. Furthermore, several in vitro studies have shown the anticarcinogenic effects of VK2, and the results from the Heidelberg cohort of the European Prospective Investigation into Cancer and Nutrition (EPIC) found that VK2 intake was inversely associated with the incidence of advanced prostate cancer [49,50,51,52,53].

## 3. Vitamin K2 and Alzheimer’s Disease

### 3.1. The Antiapoptotic and Antioxidant Effects of Vitamin K2

We will first consider the effects of VK2 and Gas6 on β-amyloid (Aβ). Aβ leads to neuronal death both by promoting apoptosis and by direct toxicity [42]. Neurotoxicity occurs through a variety of mechanisms, including the disruption of calcium homeostasis, oxidative stress, and mitochondrial dysfunction. In PC12 cells derived from a rat pheochromocytoma, VK2 prevented neuronal death resulting from Aβ(1–42), the most neuro-toxic form of Aβ [42]. When cells were exposed to either hydrogen peroxide (H_2_O_2_) or Aβ(1–42), the cells that were pretreated with VK2 exhibited markedly less apoptosis, as measured by flow cytometry (Table 1). Pretreatment with VK2 also decreased the amount of apoptosis signaling proteins, including a lower Bax/Bcl-2 ratio, reduced the presence of reactive oxygen species (ROS), and increased the amount of glutathione, a powerful anti-oxidant. The authors identified the inactivation of the p38 MAP kinase pathway as a mechanism for the potential protective role of VK2 in Alzheimer’s disease (AD).

A 2021 study reached similar conclusions in a C6 cell line of rat astroglia that was transfected in order to express Aβ. Upon an increase in the concentration of VK2, cells exhibited prolonged survival due to protection against Aβ-induced neuronal death [43]. This effect was reversible upon the addition of warfarin, which prevents vitamin K-dependent carboxylation. VK2 also reduced the number of ROS in a dose-dependent manner and, at a concentration of 10 mcmol/L, decreased by 2.5-fold the activity of caspase-3, an enzyme that mediates Aβ-induced apoptosis. Furthermore, the authors found that Gas6 plays a role in VK2 protection against Aβ cytotoxicity, supporting findings from an older study. That study, from 2002, had measured Ca^(2+)^ influx, chromatin condensation, and DNA fragmentation as markers for Aβ neurotoxicity and apoptosis in rat embryo neuronal cell cultures [54]. Gas6 inhibited the influx of Ca^(2+)^ in a dose-dependent manner and significantly decreased the amount of chromatin condensation and DNA fragmentation caused by Aβ. When considered together, these studies present a convincing argument for the antiapoptotic and antioxidant properties of VK2.

### 3.2. Vitamin K2 and Neuroinflammation

It has become increasingly clear that neuroinflammation and chronic glial hyperactivation are important in neurodegeneration and the pathogenesis of AD [55,56,57,58,59,60,61,62,63]. While astrocytes and microglia can play a neuroprotective role via the clearing of Aβ, excessive chronic activation can accelerate, or even cause, neurodegeneration [57,58,59,60,61,62]. One study in animal models showed that glial activation occurs prior to the formation of Aβ plaques, while another study suggested that neuroinflammation may initiate neuronal dysfunction (assessed by MRI scans) in AD and Parkinson’s disease [55,56]. Additionally, mutations in the microglial receptors CD33 and TREM2 have been associated with an increased risk of AD, and the proximity of AD risk loci to genes highly expressed in microglia suggests that neuroinflammation is critical to AD pathogenesis [64,65,66,67].

When microglia are activated, they initiate an inflammatory cascade, releasing an excess of proinflammatory mediators that include the cytokines tumor necrosis factor alpha (TNF-α), interleukin 1β (IL-1β), and interleukin 6 (IL-6) [57,58,59,68]. Several factors trigger microglial activation: oxidative stress, chronic infection, injury, Aβ, and lipopolysaccharide (LPS), a well-known endotoxin that is a major component of the cell wall in gram-negative bacteria [57,58,61,68,69] (Figure 4). The disruption of microglial homeostasis contributes to a persistent state of neuroinflammation and ultimately causes neurodegeneration through a variety of mechanisms: synaptic loss, neuronal death, and the activation of neurotoxic astrocytes [70,71,72].

MK-4 suppressed microglial inflammation in mouse microglia-derived MG6 cells exposed to LPS [47]. Pretreatment with MK-4 inhibited NF-κB signaling, the production of the inflammatory cytokines TNF-α, IL-1β, and IL-6, and the upregulation of cytokines at the mRNA level. Another study found that MK-7 reversed the upregulation of proinflammatory cytokines caused by glial activation in rat astrocytes [73]. Cells were cultured un-der both normal and hypoxic conditions, and those that were pretreated with MK-7 exhibited a decrease in the hypoxia-induced expression of IL-6 and TNF-α. Additionally, MK-7 suppressed ROS production in the hypoxic astrocytes, confirming the antioxidant effects of VK2 shown by [42,43]. Despite using different types of glial cells (microglia and astrocytes), distinct forms of VK2 (MK-4 and MK-7), and different methods of activation (LPS and hypoxia), these two studies both reached the conclusion that VK2 suppresses the production of proinflammatory cytokines, suggesting that VK2 may have the potential to reduce neuroinflammation and neurodegeneration.

### 3.3. Vitamin K2 and Mitochondrial Dysfunction

Mitochondrial impairment in AD has been documented by decades of research, and the connection between mitochondrial dysfunction, neuroinflammation, and neurodegeneration in AD, Parkinson’s disease, and multiple sclerosis has been described [74,75]. In the previously discussed study that exposed an astrocyte culture to hypoxia, pretreatment with MK-7 not only reduced neuroinflammation, but also increased ATP production and suppressed ROS production in hypoxic astrocytes [73]. In a different study, MK-4 rescued severe mitochondrial defects in fruit flies carrying a mutated version of the *pink1* gene, which causes familial Parkinson’s disease in humans [76]. Supplementation with MK-4 improved flight and increased ATP production in *pink1* mutants in a dose- and time-dependent manner.

When Drosophila flies transgenic for AD were treated with VK2 for 28 days, VK2 decreased brain Aβ(1–42) levels, augmented ATP production, rescued mitochondrial dysfunction, improved climbing ability, and prolonged lifespan [77]. Additionally, VK2 upregulated the expression of genes responsible for activating autophagy, the lysosomal-mediated degradation of abnormal proteins. It is notable that VK2 promotes autophagy in a variety of cell types given that activating this process has been shown to decrease Aβ neurotoxicity and improve cognition [49,50,51,52,78,79,80]. We believe that the abilities of VK2 to reduce Aβ neurotoxicity and rescue mitochondrial dysfunction make it of interest as a potential new therapy for AD.

### 3.4. Vitamin K2 and Anesthesia-Induced Cognitive Deficits

Two studies examining the effects of anesthetics on mice that either expressed Aβ or had phosphorylated tau proteins both found that VK2 raised ATP levels and mitigated cognitive impairment resulting from anesthesia. A 2018 study compared the effects of two types of anesthetics (isoflurane and desflurane) on transgenic mice and wild-type mice. The transgenic mice had mutated versions of the *APP* gene found in familial AD and, when given isoflurane, experienced cognitive delay. Analysis of their brain tissue showed that they had a decreased number of synapses and reduced levels of ATP production in the hypothalamus, negative effects that were reversed by VK2 treatment [81]. In support of these findings, a 2020 study found that VK2 mitigated tau phosphorylation and cognitive deficits induced by sevoflurane in newborn mice [82]. Given that these studies looked at different markers for AD in mice, the fact that they both provide evidence of the neuroprotective effects of VK2 suggests that VK2 might play an important role in AD treatment.

### 3.5. Vitamin K2 and Cardiovascular Health

Numerous population studies have reported that atherosclerosis, arterial calcification, and arterial stiffness increase the risk for dementia and cognitive impairment [83,84,85,86,87,88,89,90,91]. In a population-based cohort study of 844 patients, the amount of atherosclerotic calcification was directly proportional to cognitive impairment and inversely related to brain tissue volume [83]. Furthermore, a greater volume of calcification, as measured by a computed tomography (CT) scan, corresponded to reduced microstructural integrity of white matter. Three years later, a large population study that followed 2364 participants for five years found that atherosclerotic calcification was associated with an increased risk of developing dementia [84]. A cohort study with 1732 participants reported a similar finding and established an association between generalized atherosclerosis and mild cognitive impairment [85]. One study demonstrated that carotid artery calcification was associated with a higher risk of dementia, and a recent cohort study that followed 4988 middle-aged Dutch ImaLife participants over a 10-year period showed that coronary artery calcium severity was associated with cognitive decline [86,87]. Additionally, aortic stiffness has been associated with an increased risk of dementia [88].

Compelling evidence also suggests that vascular health is closely linked to AD. Cerebral atherosclerosis, small vessel disease, cerebral amyloid angiopathy, and blood-brain barrier dysfunction have all been reported in AD [92]. Arterial stiffness, arteriolosclerosis, and endothelial dysfunction are characteristic of cerebrovascular disease, which contributes to neurovascular disintegration, brain atrophy, and the accumulation of cerebral Aβ [93,94]. Cardiovascular disease (CVD) has been shown to play an important role in the etiology of AD, and epidemiological studies have established that vascular disease increases AD risk [89,90,91,95,96,97,98]. Interventions improving vascular function also have attenuated AD pathology [93]. One study of 2907 elderly individuals demonstrated that the *APOE* gene serves as a genetic link between AD and vascular disease [99].

In addition, pathological data have suggested that vascular and neurodegenerative processes can coexist in AD [100]. Autopsies of individuals with dementia revealed that 80% of cases had AD, 7–10% had vascular dementia, and 3–5% had mixed dementia, with vascular lesions being observed in 20–40% of the subjects who had AD [101]. Vascular lesions included cortical microinfarcts, white matter lesions, small hemorrhages, and corticosubcortical infarcts. The vascular lesions in subjects who had AD were typically smaller than the lesions in subjects who had either vascular or mixed dementia. Another study involving autopsies found an association between vascular risk factors, cerebrovascular disease, and AD among 5715 subjects [102].

Now, we will review studies that have highlighted the role of VK2 in vascular health. In a study that followed 4807 men and women of age 55 and above for more than seven years, the participants in the highest tercile for VK2 intake had a 41% lower risk of CVD when compared with the those in the lowest tercile [103]. VK1 had no effect on CVD risk. A more recent, 11-year study of 2987 Norwegian adults between the ages of 46 and 49 found a 50% decrease in CVD risk in individuals with the highest levels (fourth quartile) of VK2 dietary intake [104].

Two different types of studies conducted on women also came to the conclusion that VK2, and not VK1, decreases the risk of CVD. In a population-based cohort study that followed 16057 women between 49 and 70 years of age for a mean duration of eight years, the risk of developing coronary heart disease (CHD), a type of CVD, was inversely proportional to VK2 intake [105]. The authors identified a hazard ratio of 0.91 per 10 mcg of VK2 intake, indicating that each increase of 10 mcg of VK2 lowered the risk of CHD by an average of 9%. We note that they attribute the inverse association between VK2 intake and the risk of CHD to MK-7, MK-8, and MK-9 in particular. In a cross-sectional study of 564 postmenopausal women, a higher dietary intake of VK2 led to a decrease in coronary calcification, a marker for CHD [106].

Another study of postmenopausal women strengthens the argument for the benefit of VK2 in cardiovascular health. A double-blind, placebo-controlled trial that followed 244 healthy postmenopausal women over three years demonstrated that supplementation with 180 mcg MK-7 improved arterial stiffness, especially in individuals who had a lower baseline arterial elasticity [107]. Interestingly, a much shorter trial that followed 68 men and women came to a different conclusion [108]. Participants were given a 360-mcg daily supplement of MK-7, and CT scans after six months showed no change in the amount of arterial calcification in patients with type II diabetes or CVD. However, we believe that six months may not have been sufficient time to allow for a detectable change in calcification.

There are two studies that not only disagree with the general consensus of the aforementioned studies, but that also contradict each other. A large population study found a borderline significant association between VK2 intake and CVD mortality (acknowledging that there were only a small number of CVD-related deaths) and concluded that VK1 and VK2 do not impact overall mortality [109]. A study of a Mediterranean population reached the opposite conclusion—both VK1 and VK2 decreased all-cause mortality and cancer, but only VK1 was associated with decreased CVD risk [110]. Nonetheless, the study’s authors concluded that both VK1 and VK2 have a potential protective role in cardiovascular mortality, cancer mortality, and all-cause mortality. These two studies, which we include in this review for thoroughness, are not that pertinent to our investigation because of the presence of confounding factors and their focus on cancer mortality and CVD mortality rather than CVD events.

Two studies done in rodents support the beneficial effects of VK2 on CVD risk. In rats, MK-4, but not VK1, prevented arterial calcification induced by warfarin [111]. In transgenic ApoE/LDLR−/− mice with atherosclerotic plaques, MK-7 treatment improved endothelial function [112]. We believe this finding is important given the fact that dysfunction of the endothelium at the level of the blood-brain barrier has been found to contribute to the development of AD [113].

Although the studies we have reviewed took different approaches, most came to the conclusion that VK2, and not VK1, has a beneficial role in arterial health. Given the connection between vascular health and AD that we have highlighted, these findings suggest that VK2 might be relevant to AD prevention.

### 3.6. Vitamin K2 and the Gut Microbiome

An emerging area of research is the role of the gut microbiome in brain health. We will first discuss four recent studies and one case report that all involve fecal microbiota transplantation (FMT) and suggest that gut microbiome changes contribute to AD pathogenesis. In APPswe/PS1dE9 mice, which are transgenic for AD, improving microbiome composition through FMT from healthy, wild-type mice increased synaptic plasticity and decreased the amount of Aβ plaques [114]. A study of 5× FAD transgenic mice found similar results [115]. In support of previous findings, FMT improved cognition and reduced the formation of both Aβ plaques and tau protein tangles in ADLPAPT transgenic mice [116]. Furthermore, the authors noted an improvement in immune function and a decrease in glial reactivity. Despite taking a different approach, another study also reached the conclusion that the microbiome plays an essential role in AD pathogenesis [117]. One group of seven germ-free mice was transplanted with fecal microbiota from a healthy 76-year-old female donor, and a second group was transplanted with fecal microbiota from an 82-year-old male with AD. The mice in the AD group exhibited cognitive decline, a decrease in the neurotransmitter gamma-aminobutyric acid (GABA), and a reduction in the levels of taurine and valine, amino acids that are important for nervous system function. A case report published last year described the results when an 82-year-old male with AD underwent an FMT for an antibiotic-resistant *Clostridium difficile* (*C. diff*) infection [118]. The patient’s cognition improved in subsequent months, as demonstrated by an increase in his mini-mental state exam (MMSE) score from 20 to 29 over a four-month period. (An MMSE score of 30 indicates normal cognitive function.) We believe that the above studies strongly suggest a causal relationship between dysbiosis and AD pathogenesis.

There are also other studies that document a connection between gut microbiome composition and AD. In APPswe/PSEN1dE9 mice, gut microbiome disruption using antibiotics impacted innate immunity, neuroinflammation, and Aβ amyloidosis [119]. Another study of transgenic APP mice documented significant differences between intestinal microbiome profiles of transgenic and wild-type mice [120]. Furthermore, in the transgenic mice, the complete eradication of the gut microbiome led to a reduction in cerebral Aβ. In both mice and humans, the APOE genotype has been associated with unique gut microbiome profiles [121]. When rats exhibiting symptoms characteristic of AD were given the prebiotic fructooligosaccarides, their cognition improved [122]. A randomized control trial (RCT) with 60 AD patients who received a probiotic containing a mix of the bacteria *Lactobacillus acidophilus*, *Lactobacillus casei*, *Bifidobacterium bifidum*, and *Lactobacillus fermentum* for 12 weeks reached a similar conclusion [123]. Another RCT found that a modified Mediterranean-ketogenic diet improved gut microbiome composition and AD biomarkers in cerebrospinal fluid [124]. An observational study in humans also showed that gut microbiome changes were associated with cerebrospinal fluid biomarkers of AD [125]. Many studies done in animal models support the argument that the gut microbiome is altered in AD [122,126,127,128,129,130,131]. The gut microbiome can also play a role in neuroinflammation; for example, pathogenic gut bacteria have been shown to trigger chronic glial hyperactivation, blood-brain barrier dysfunction, and increased intestinal barrier permeability [132,133,134].

However, two questions that remain unanswered are how dysbiosis affects VK2 production and what impact this change has on human health. To convey our argument for the bioavailability of intestinal VK2 and its importance for human heath, we will analyze the two studies that suggest that VK2 synthesized by gut bacteria cannot be absorbed. One influential study from 1992 demonstrated that MK-4, but not MK-9, can be absorbed from the colon [135]. Based on the observations from this study and given the fact that the majority of the gut bacteria reside in the colon (large intestine), it was proposed that long-chain MKs do not contribute to the total human physiological requirement of VK2. Although often overlooked, one notable finding of the study is that MK-9 can be absorbed from the ileum (small intestine).

In fact, there have been several studies that demonstrate that bacterially-synthesized MKs are present in the ileum. While there are far less bacteria in the ileum than in the large bowel, the number of bacteria in the ileum can still explain the presence of MKs in the small bowel. A small portion of MKs is present in the distal small intestine, where they can be absorbed in the presence of bile salts [136]. A follow-up study reported a larger amount of MK-9 and MK-10 compared to MK-4 and MK-7 in both the terminal ileum (7.93 versus 0.92) and the ileostomy (1.16 versus 0.69) [137]. Reinforcing the above findings, one study suggests that the amount of VK2 synthesized in the gut by far exceeds human nutritional requirements, even if only a small fraction is absorbed [138]. Another argument for the intestinal absorption of VK2 is that the human liver contains large amounts of the long-chain MKs, which are not present in significant amounts in the diet [139].

A second study that is often cited as an argument against the biological activity of intestinally-produced VK2 is an infant study comparing formula-fed and breastfed infants that found no absorption in the former but, inexplicably, MK-7 absorption in the latter [140]. Because a newborn’s microbiome at one week of age is undeveloped, we would argue that these results should be not be extrapolated to adults. Furthermore, we note that a study from the 1990s urged caution in interpreting published concentrations of VK2, due not only to the technical limitations that hamper the ability to accurately measure small concentrations of serum MKs, but also due to the wide variability in those values as reported by different researchers [138].

By demonstrating that microbiome disruption in small intestinal bacterial overgrowth (SIBO) or following antibiotic treatment decreases VK2 production and disturbs body homeostasis, several studies provide a compelling argument for the absorption of intestinal VK2. A 2019 study revealed that antibiotic-triggered disruption of the mice microbiome decreased not only bone strength and bone density, but also the number of microbial genes involved in MK synthesis [141]. As a result, there was also a decrease in the amount of VK2 in the cecum, liver, and kidney. These findings are supported by a study that showed that antibiotic treatment in mice fed a diet low in vitamin K eradicated the gut bacteria that produce menaquinone, leading to vitamin K deficiency and gastric hemorrhage [142]. 16S rRNA sequencing confirmed a decrease in the *menA* and *menD* genes, which are involved in VK2 biosynthesis. In another study, decreased levels of VK2 in SIBO were associated with increased arterial calcification and subclinical atherosclerosis [143]. In a study that investigated the connection between cognition and bacteriallysynthesized VK2, researchers measured the fecal menaquinone levels of 74 elderly individuals, and the MK-6, MK-12, and MK-13 forms were positively associated with cognition [144].

As the link between the disruption of the gut microbiome and AD pathogenesis has become increasingly clear, we hypothesize that changes in microbiome composition would alter VK2 production and the likelihood of developing AD. Therefore, we believe that it is imperative to further explore the role of dysbiosis and VK2 production in AD.

### 3.7. Vitamin K2 and Comorbidities in Alzheimer’s Disease

Type II diabetes is a risk factor for and a comorbidity in AD. Three different RCTs found that supplementation with VK2 reduced glycemia and insulin resistance in patients with type II diabetes [145,146,147]. VK2 intake has been shown to be inversely associated with the risk of type II diabetes [148]. A cohort study investigating the effect of VK2 on metabolic syndrome (MetS), a condition associated with hyperglycemia and an increased risk of diabetes, found an inverse association between VK2 intake and the occurrence of MetS [149]. In another study, an analysis of the gut microbiome of 12 individuals with type II diabetes and six healthy individuals through shotgun metagenomic sequencing revealed that metabolic pathways involved in VK2 biosynthesis were enriched in the subjects with type II diabetes [150].

In animal models, osteocalcin, an extrahepatic VKDP that is activated by VK2, has reduced insulin resistance [36,151]. Other studies support the argument that osteocalcin might play an important role in type II diabetes [147,152,153]. Osteoporosis is frequently found in AD patients, and there is an increased risk of fractures in individuals with dementia [154,155,156]. In fact, patients with osteoporosis are at a higher risk of developing AD [157,158,159]. In the context of these findings, it was suggested that there is a possible link between osteoporosis and AD [156]. The role of VK2 in osteoporosis treatment and prevention has been documented by clinical trials and research studies, and VK2 has been approved for the treatment of osteoporosis in Japan [48,160,161,162,163,164,165]. We believe that the importance of VK2 in bone health is yet another reason for clinical trials with VK2 in individuals with AD.

Depression is both a risk factor for AD and a comorbid condition that worsens the prognosis of AD [166,167,168]. Depression can precede dementia and usually occurs in up to 50% of AD patients [168]. One study in rats that had MetS demonstrated that VK2 prevents the development of anxiety and depression [169].

## 4. Discussion

VK2 has recently emerged as an essential nutrient for human health, and a growing number of studies have demonstrated that VK2 and vitamin K-dependent proteins (VKDPs) may play an important role in slowing, and even preventing, the progression of AD. VK2 improves neuronal health through a variety of mechanisms: reducing apoptosis induced by β-amyloid (Aβ), limiting oxidative stress, reversing microglial activation, suppressing neuroinflammation, and improving vascular health.

Because large pharmacological trials have failed to find an effective medication to treat or prevent AD, there has been a shift in AD research to nonpharmacological interventions. As of February 2021, there were 270 clinical trials funded by the National Institute of Aging that investigated AD and related dementias, and the number of nonpharmacological studies (123) was more than double the number of pharmacological ones (57) [170]. Despite the fact that the relationship between VK2 and other diseases, such as osteoporosis and cardiovascular disease, has been studied, there is a lack of clinical trials investigating the connection between VK2 and AD. Out of the 121 clinical studies with VK2 in humans, none explore the connection between VK2 and AD [171].

The absence of human clinical studies is especially puzzling considering that VK2 is associated with the progression of multiple sclerosis (MS) and Parkinson’s disease (PD), which both are neurodegenerative diseases. An observational study of 45 MS patients and 29 healthy controls demonstrated that serum concentrations of VK2 decreased by more than three-fold in MS patients and that, among those with MS, lower levels of VK2 were associated with a higher frequency of MS attacks [172]. A similar study of 93 PD patients and 95 healthy controls showed that serum levels of VK2 were reduced in PD and that, among the PD patients, VK2 levels were lowest in those with late-stage PD [173].

This paper has reviewed studies that consider the relationship between VK2 and Aβ neurotoxicity, neuroinflammation, mitochondrial dysfunction, cognition, cardiovascular health, dysbiosis, and AD comorbidities. When considered together, these studies strongly suggest that VK2 could play an important role in AD prevention and therapy. Because of the increasing evidence for a link between VK2 and AD, we were surprised to learn that there have not yet been any clinical studies in humans that investigate this connection. In the context of the paradigm shift in AD research toward nonpharmacological interventions, and to fill the gap in clinical research, we argue that it is of critical importance to investigate any possible connections between VK2 levels and AD risk through observational studies and RCTs.

## Figures and Tables

**Figure 1 nutrients-13-02206-f001:**
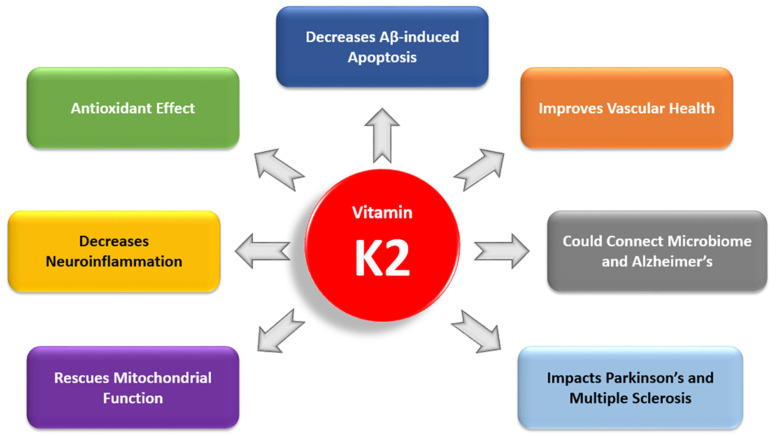
The roles of vitamin K2 in Alzheimer’s disease.

**Figure 2 nutrients-13-02206-f002:**
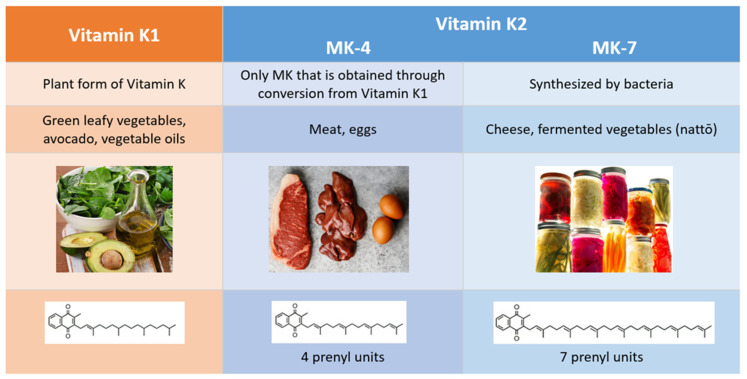
Differences between vitamin K1 and vitamin K2.

**Figure 3 nutrients-13-02206-f003:**
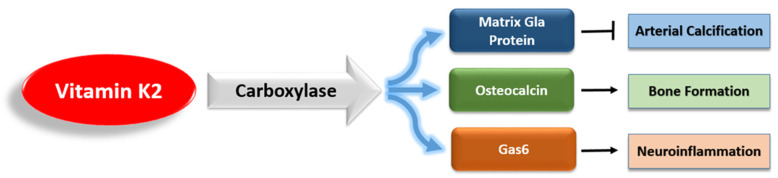
The main vitamin K-dependent proteins that are activated by vitamin K2.

**Figure 4 nutrients-13-02206-f004:**
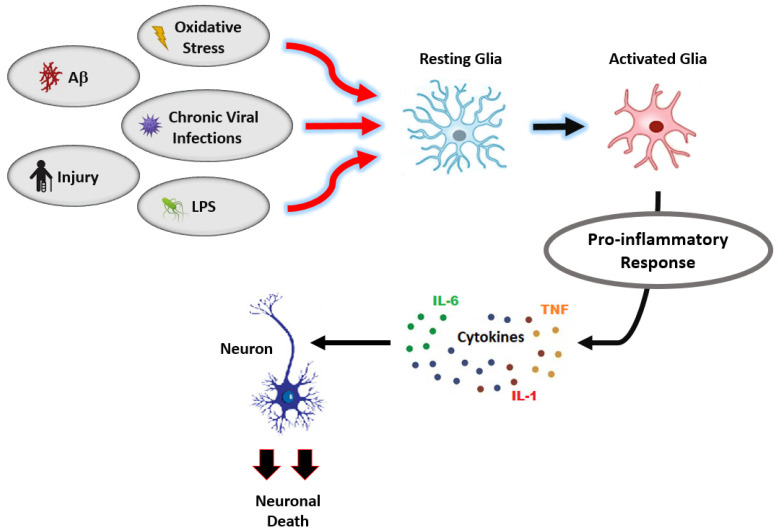
Glial activation leads to neurodegeneration.

**Table 1 nutrients-13-02206-t001:** The effects of vitamin K2 on Aβ(1–42) and H_2_O_2_ induced apoptosis, as described by [42].

Experimental Group	Percent of Apoptotic Cells (%)
Control for Aβ(1–42)	3.2
20 mcmol K2 + Aβ(1–42)	4.3
Aβ(1–42) Only	13.1
Control for H_2_O_2_	4.7
20 mcmol K2 + H_2_O_2_	6.4
H_2_O_2_ Only	28

## Data Availability

Not applicable.

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
