# Peer review of "Vitamin K2 Holds Promise for Alzheimer’s Prevention and Treatment"

_nutrients, 2021, doi:10.3390/nu13072206_

Round 1

Reviewer 1 Report

Dear Authors,

This is very important review in my opinion. Vitamin K is mainly recognized as coagulation vitamin. We can find new research around vitamin K2 and bone and cardiovascular health but there is not so much articles, which are focused on brain, mental health and the importance of vitamin K2.  I really appreciated reading your review.

Some issues:

  1. It seems important to include information about protein S (VKDP) . As for vascular dementia, the main causes are represented by several vascular pathologies that result in cerebral ischemia. Studies published in the last years have attributed to protein S (activated by vitamin K) a role in improving post-ischemic cerebral blood flow and potentially leading to a more favorable cognitive outcome. Ref Liu D, Guo H, Griffin JH, Fernández JA, Zlokovic BV. Protein S confers neuronal protection during ischemic/hypoxic injury in mice. Circulation. 2003;107(13):1791-1796.

  1. You showed the connection between Alzheimer and CV health, but also osteoporosis is connected with AD, and Vitamin K2 is connected with bone health. Vitamin K2 as MK4 is a drug in Japan to treat osteoporosis. MK7 was shown to improve bone mineral density and strength in some clinical studies too.

Previous observational studies have reported the increased frequency of comorbid osteoporosis in AD. The relationship between these two diseases is more likely one of shared etiology than one condition causing the other. There are two major pathophysiological links for osteoporosis and AD. The overexpression of Aβ may take place in both brain and bone, interfering with the RANKL signaling cascade, enhancing osteoclast activities, and leading to osteoporosis. “ ref https://www.ncbi.nlm.nih.gov/pmc/articles/PMC5615992/

So patients with AD might benefit from K2 supplementation from bone perspective.

  1. Depression and AD. Depression is a comorbid condition in Alzheimer's disease with negative consequences in patients and caregivers. Depression may predate dementia and tends to occur in up to 50% of AD patients with a decrease of noradrenalin and serotonin in the brain being the most plausible cause ref https://pubmed.ncbi.nlm.nih.gov/21504132/

Vitamin K2 was evaluated only in animal study so far.

In an animal study researchers examined the effects of vitamin K2 on the behavior of rats with metabolic syndrome and looked for relationships with the effects on blood sugar. This study demonstrated that vitamin K2 prevented the development of anxiety and depression, but did not improve the memory deficit caused by the dietary manipulation in an experimental model of metabolic syndrome. Ref https://europepmc.org/article/med/28068285

However based on the human study we know that that depressive symptoms were significantly lower in people with higher dietary vitamin K intake.

https://pubmed.ncbi.nlm.nih.gov/30959758/

  1. Moreover the use of VKAs was shown to influence brain metabolism. Maye it is important to mention this fact too.

 The few papers published until now point out, to a limited extent, a potential correlation between the use of VKAs and both cognitive decline and brain focal atrophies. Example of ref https://pubmed.ncbi.nlm.nih.gov/25151653/

https://pubmed.ncbi.nlm.nih.gov/26576841/

https://pubmed.ncbi.nlm.nih.gov/29486479/

Author Response

Dear Authors, 

This is very important review in my opinion. Vitamin K is mainly recognized as coagulation vitamin. We can find new research around vitamin K2 and bone and cardiovascular health but there is not so much articles, which are focused on brain, mental health and the importance of vitamin K2.  I really appreciated reading your review.

Thank you for taking the time to review and comment upon our manuscript, nutrients-1269058, “Vitamin K2 Holds Promise for Alzheimer’s Prevention and Treatment.” We found the advice constructive and have incorporated your suggestions into our revision.

We’ve responded to each comment individually below (see italics).

Thank you again for your thoughtful comments.

Kind regards,

Alexander Popescu and Monica German, MD

Some issues:

  1. It seems important to include information about protein S (VKDP) . As for vascular dementia, the main causes are represented by several vascular pathologies that result in cerebral ischemia. Studies published in the last years have attributed to protein S (activated by vitamin K) a role in improving post-ischemic cerebral blood flow and potentially leading to a more favorable cognitive outcome. Ref Liu D, Guo H, Griffin JH, Fernández JA, Zlokovic BV. Protein S confers neuronal protection during ischemic/hypoxic injury in mice. Circulation. 2003;107(13):1791-1796.

Thank you for the recommendation. We have added information about protein S to section 2 of the manuscript (Comparison of VK1 and VK2).

  1. You showed the connection between Alzheimer and CV health, but also osteoporosis is connected with AD, and Vitamin K2 is connected with bone health. Vitamin K2 as MK4 is a drug in Japan to treat osteoporosis. MK7 was shown to improve bone mineral density and strength in some clinical studies too.

“Previous observational studies have reported the increased frequency of comorbid osteoporosis in AD. The relationship between these two diseases is more likely one of shared etiology than one condition causing the other. There are two major pathophysiological links for osteoporosis and AD. The overexpression of Aβ may take place in both brain and bone, interfering with the RANKL signaling cascade, enhancing osteoclast activities, and leading to osteoporosis. “ ref https://www.ncbi.nlm.nih.gov/pmc/articles/PMC5615992/

So patients with AD might benefit from K2 supplementation from bone perspective.

Thank you for the recommendation. We have added a new section, “3.7 Vitamin K2 and comorbidities in Alzheimer’s disease” that includes information about osteoporosis and AD.

  1. Depression and AD. Depression is a comorbid condition in Alzheimer's disease with negative consequences in patients and caregivers. Depression may predate dementia and tends to occur in up to 50% of AD patients with a decrease of noradrenalin and serotonin in the brain being the most plausible cause ref https://pubmed.ncbi.nlm.nih.gov/21504132/

Vitamin K2 was evaluated only in animal study so far.

In an animal study researchers examined the effects of vitamin K2 on the behavior of rats with metabolic syndrome and looked for relationships with the effects on blood sugar. This study demonstrated that vitamin K2 prevented the development of anxiety and depression, but did not improve the memory deficit caused by the dietary manipulation in an experimental model of metabolic syndrome. Ref https://europepmc.org/article/med/28068285 

However based on the human study we know that that depressive symptoms were significantly lower in people with higher dietary vitamin K intake.

https://pubmed.ncbi.nlm.nih.gov/30959758/

Thank you for the recommendation. We have added a new section, “3.7 Vitamin K2 and comorbidities in Alzheimer’s disease” that includes information about depression and AD.

  1. Moreover the use of VKAs was shown to influence brain metabolism. Maye it is important to mention this fact too.

The few papers published until now point out, to a limited extent, a potential correlation between the use of VKAs and both cognitive decline and brain focal atrophies. Example of ref https://pubmed.ncbi.nlm.nih.gov/25151653/

https://pubmed.ncbi.nlm.nih.gov/26576841/

https://pubmed.ncbi.nlm.nih.gov/29486479/

Thank you for the recommendation. We have added information about VKAs to section 2 of the manuscript (Comparison of VK1 and VK2).

Reviewer 2 Report

Drs. Popescu and German present a study reviewing the recent investigations of the role and function of vitamin K2 on Alzheimer’s disease.

In this review article, the authors briefly introduced vitamin K, both K1 and K2. They summarized the possible function and mechanism of vitamin K2 in the prevention of AD in several aspects. These aspects include anti-apoptosis, anti-oxidation, anti-inflammation, mitochondrial dysfunction etc.. Basically, this review article covers most aspects and provides a good interest for future research to develop the possible application of vitamin K2 on the prevention and possible treatment of neurodegenerative diseases.

Author Response

Drs. Popescu and German present a study reviewing the recent investigations of the role and function of vitamin K2 on Alzheimer’s disease.

In this review article, the authors briefly introduced vitamin K, both K1 and K2. They summarized the possible function and mechanism of vitamin K2 in the prevention of AD in several aspects. These aspects include anti-apoptosis, anti-oxidation, anti-inflammation, mitochondrial dysfunction etc.. Basically, this review article covers most aspects and provides a good interest for future research to develop the possible application of vitamin K2 on the prevention and possible treatment of neurodegenerative diseases.

Thank you for taking the time to review and comment upon our manuscript, nutrients-1269058, “Vitamin K2 Holds Promise for Alzheimer’s Prevention and Treatment.”

Kind regards,

Alexander Popescu and Monica German, MD

Reviewer 3 Report

This article aims to review the basic science and clinical studies that look at Vitamin K2 (VK2) and factors that contribute to Alzheimer’s disease (AD pathogenesis). The manuscript will consider “antiapoptotic and antioxidant effects of VK2 and its impact on neuroinflammation, mitochondrial dysfunction, cognition, and cardiovascular health. We also examine the link between dysbiosis and VK2 in the context of the microbiome’s role in AD pathogenesis.”

Significance:

There are other small reviews on VK2 and neurodegeneration and dementia. This is the first review to focus on VK2 and AD. This seems to be the only review that comments on VK2 and AD from my quick search.

The introduction is well laid out with a discussion of AD and a comparison of VK1 and VK2. The review nicely lays out the connection between VK2 and components of AD including neuroinflammation, mitochondrial dysfunction, cognition, and cardiovascular health.

The discussion of AD nicely lays out the relationship of AB, tau, and genetic factors. A short discussion of environmental factors that contribute to AD would be beneficial.

There is a very clear and concise comparison of VK1 and VK2 (MK-4 and MK-7). After this, the authors jump right into VK2 and AD. I understand why they are doing that, but some sort of short transition talking about VK1 and VK2 and AD would beneficial, including some references for VK1 and AD.

Section 3.

Section 3.1

This sentence needs a reference “Aβ leads to neuronal death both by promoting apoptosis and by direct toxicity.”

Section 3.2

Also needs a reference: “In recent years, there has been increasing evidence of the importance of microglial activation and neuroinflammation in AD pathogenesis.”

Also needs a reference or references: “When microglia are activated, they initiate an inflammatory cascade that releases an excess of proinflammatory media-tors that include the cytokines tumor necrosis factor alpha (TNF-α), interleukin 1 alpha (IL-1α), interleukin 1 beta (IL-1β), and interleukin 6 (IL-6).”

Section 3.4

“Given that these studies looked at markers for AD in mice, the fact that they both provide evidence of the neuroprotective effects of VK2 bolsters the argument that VK2 plays an important role in AD treatment.” This is overstated. In my opinion, these studies only hint that VK2 might be important.

Section 3.5

“Compelling evidence suggests a connection between cardiovascular disease and AD pathogenesis, with numerous population studies reporting that atherosclerosis, arterial calcification, and arterial stiffness increase the risk for dementia and cognitive impairment.” Need references.

Agree with the following sentence “We believe that these studies establish a convincing association between vascular health and impaired cognition.” Can you make a strong tie to AD as opposed to vascular dementia?

I am actually not fully convinced of the following “Although the studies we have reviewed took different approaches, most came to the conclusion that only VK2 impacts arterial health, helping to reinforce the claim that VK2 has a beneficial role in cardiovascular health and thus AD pathogenesis.”  It is pretty clear that VK2 plays a role in vascular health, but I am not convinced it alters AD.  I would suggest revising and adding more references if that is possible.

Section 3.6

“An emerging area of research is the role of the gut microbiome in brain health. While there are hundreds of studies discussing the connection between the gut microbiome and AD, we highlight four recent studies and one case report that all involve fecal microbiota transplant (FMT) and suggest that gut microbiome changes contribute to AD pathogenesis.” Why are you only looking at these? It would be beneficial to let the reader know why.

“To convey our argument for the bioavailability of intestinal VK2 and its importance for human heath, we will analyze two key studies that suggest that VK2 synthesized by gut bacteria cannot be absorbed.“ Why? It seems like you are picking and choosing data to share.

“Furthermore, we note that a study from the 1990s urged caution in interpreting published concentrations of VK2, due not only to the technical limitations that hamper the ability to accurately measure small concentrations of serum MKs, but also due to the wide variability in those values as reported by different researchers [72].” Has this changed since the 90s?

“As the link between the disruption of the gut microbiome and AD pathogenesis has become increasingly clear..” I actually don’t think it’s that clear from what you presented.  You had a very narrow focus in this section.

Overall, this is a solid review article highlighting the potential preventative and/or therapeutic potential of VK2 in AD. The review seems very limited in scope. While some studies may be lacking, I don’t understand why the section on the gut microbiome is so limited. The review is also very skewed. There seems to be a multitude of evidence (maybe?) on VK2 and the microbiome and AD, but the rest of the evidence (inflammation?) seems very thin. Maybe framing the review would be better? It’s good information and needed, but some changes would make this much better.

Author Response

This article aims to review the basic science and clinical studies that look at Vitamin K2 (VK2) and factors that contribute to Alzheimer’s disease (AD pathogenesis). The manuscript will consider “antiapoptotic and antioxidant effects of VK2 and its impact on neuroinflammation, mitochondrial dysfunction, cognition, and cardiovascular health. We also examine the link between dysbiosis and VK2 in the context of the microbiome’s role in AD pathogenesis.” 

Thank you for taking the time to review and comment upon our manuscript, nutrients-1269058, “Vitamin K2 Holds Promise for Alzheimer’s Prevention and Treatment.” We found the advice constructive and have incorporated your suggestions into our revision.

We’ve responded to each comment individually below (see italics). We have significantly expanded the information about VK1, neuroinflammation, microbiome, and vascular health, and we have added additional references and a new section titled “Vitamin K2 and comorbidities in Alzheimer’s disease.”

Thank you again for your thoughtful comments.

Kind regards,

Alexander Popescu and Monica German, MD

Significance:

There are other small reviews on VK2 and neurodegeneration and dementia. This is the first review to focus on VK2 and AD. This seems to be the only review that comments on VK2 and AD from my quick search. 

The introduction is well laid out with a discussion of AD and a comparison of VK1 and VK2. The review nicely lays out the connection between VK2 and components of AD including neuroinflammation, mitochondrial dysfunction, cognition, and cardiovascular health. 

The discussion of AD nicely lays out the relationship of AB, tau, and genetic factors. A short discussion of environmental factors that contribute to AD would be beneficial.

Thank you for the recommendation. We have added information about environmental factors to the introduction section of the manuscript.

There is a very clear and concise comparison of VK1 and VK2 (MK-4 and MK-7). After this, the authors jump right into VK2 and AD. I understand why they are doing that, but some sort of short transition talking about VK1 and VK2 and AD would beneficial, including some references for VK1 and AD.

Thank you for the recommendation. We have added information (including references) about VK1 and cognition/brain health to section 2 of the manuscript (Comparison of VK1 and VK2). There are limited studies specifically with VK1 and AD; however, we have included them. 

Section 3. 

Section 3.1

This sentence needs a reference “Aβ leads to neuronal death both by promoting apoptosis and by direct toxicity.”

Thank you for the recommendation. We have added a reference.

Section 3.2

Also needs a reference: “In recent years, there has been increasing evidence of the importance of microglial activation and neuroinflammation in AD pathogenesis.”

Thank you for the recommendation. We have added references. 

Also needs a reference or references: “When microglia are activated, they initiate an inflammatory cascade that releases an excess of proinflammatory media-tors that include the cytokines tumor necrosis factor alpha (TNF-α), interleukin 1 alpha (IL-1α), interleukin 1 beta (IL-1β), and interleukin 6 (IL-6).”

Thank you for the recommendation. We have added references.

Section 3.4

“Given that these studies looked at markers for AD in mice, the fact that they both provide evidence of the neuroprotective effects of VK2 bolsters the argument that VK2 plays an important role in AD treatment.” This is overstated. In my opinion, these studies only hint that VK2 might be important.

Thank you for the recommendation. We have rephrased it.

Section 3.5

“Compelling evidence suggests a connection between cardiovascular disease and AD pathogenesis, with numerous population studies reporting that atherosclerosis, arterial calcification, and arterial stiffness increase the risk for dementia and cognitive impairment.” Need references.

Thank you for the recommendation. We have added references.

Agree with the following sentence “We believe that these studies establish a convincing association between vascular health and impaired cognition.” Can you make a strong tie to AD as opposed to vascular dementia?

Thank you for the recommendation. We have added additional information and references regarding the connection between cardiovascular health and AD. We also added information on cerebrovascular pathology and AD and other types of dementia.

I am actually not fully convinced of the following “Although the studies we have reviewed took different approaches, most came to the conclusion that only VK2 impacts arterial health, helping to reinforce the claim that VK2 has a beneficial role in cardiovascular health and thus AD pathogenesis.”  It is pretty clear that VK2 plays a role in vascular health, but I am not convinced it alters AD.  I would suggest revising and adding more references if that is possible.

Thank you for the recommendation. We have added additional information and references regarding the connection between cardiovascular health and AD.

 Section 3.6 

“An emerging area of research is the role of the gut microbiome in brain health. While there are hundreds of studies discussing the connection between the gut microbiome and AD, we highlight four recent studies and one case report that all involve fecal microbiota transplant (FMT) and suggest that gut microbiome changes contribute to AD pathogenesis.” Why are you only looking at these? It would be beneficial to let the reader know why.

Thank you for the recommendation. We have expanded the section by adding additional information and references about the gut microbiome and AD.

“To convey our argument for the bioavailability of intestinal VK2 and its importance for human heath, we will analyze two key studies that suggest that VK2 synthesized by gut bacteria cannot be absorbed.“ Why? It seems like you are picking and choosing data to share.

The reason why only these two studies are mentioned is due to these studies being the only ones that we found suggesting that VK2 synthesized by gut bacteria cannot be absorbed. We have rephrased the paragraph to make this clearer. There are numerous review articles that cite these same 90’s studies.

“Furthermore, we note that a study from the 1990s urged caution in interpreting published concentrations of VK2, due not only to the technical limitations that hamper the ability to accurately measure small concentrations of serum MKs, but also due to the wide variability in those values as reported by different researchers [72].” Has this changed since the 90s?

We have not found recent studies investigating the validity of menaquinone levels. Additionally, there are very few studies that directly measure menaquinone (one used double antibody sandwich ELISA); the preferred method is to look at the ratio of carboxylated osteocalcin to uncarboxylated osteocalcin. So, yes, the technique has changed.

“As the link between the disruption of the gut microbiome and AD pathogenesis has become increasingly clear..” I actually don’t think it’s that clear from what you presented.  You had a very narrow focus in this section.

Thank you for the comment. We have expanded the section by adding additional information and references about the gut microbiome and AD. 

Overall, this is a solid review article highlighting the potential preventative and/or therapeutic potential of VK2 in AD. The review seems very limited in scope. While some studies may be lacking, I don’t understand why the section on the gut microbiome is so limited. The review is also very skewed. There seems to be a multitude of evidence (maybe?) on VK2 and the microbiome and AD, but the rest of the evidence (inflammation?) seems very thin. Maybe framing the review would be better? It’s good information and needed, but some changes would make this much better.